# ADDING GRADIENT NOISE IMPROVES LEARNING FOR VERY DEEP NETWORKS

**Arvind Neelakantan**[*][†]**, Luke Vilnis**[*][†]
College of Information and Computer Sciences
University of Massachusetts Amherst
{arvind,luke}@cs.umass.edu

**Quoc V. Le, Lukasz Kaiser, Karol Kurach**
Google Brain
{qvl,lukaszkaiser,kkurach}@google.com

**Ilya Sutskever**[†]
OpenAI
{ilyasu}@openai.com

**James Martens**
University of Toronto
{jmartens}@cs.toronto.edu

## ABSTRACT

Deep feedforward and recurrent networks have achieved impressive results in many perception and language processing applications. Recently, more complex architectures such as Neural Turing Machines and Memory Networks have been proposed for tasks including question answering and general computation, creating a new set of optimization challenges. In this paper, we explore the low-overhead and easy-to-implement optimization technique of adding annealed Gaussian noise to the gradient, which we find surprisingly effective when training these very deep architectures. Unlike classical weight noise, gradient noise injection is complementary to advanced stochastic optimization algorithms such as Adam and AdaGrad. The technique not only helps to avoid overfitting, but also can result in lower training loss. We see consistent improvements in performance across an array of complex models, including state-of-the-art deep networks for question answering and algorithm learning. We observe that this optimization strategy allows a fully-connected 20-layer deep network to escape a bad initialization with standard stochastic gradient descent. We encourage further application of this technique to additional modern neural architectures.

## 1 INTRODUCTION

Deep neural networks have shown remarkable success in diverse domains including image recognition (Krizhevsky et al., 2012), speech recognition (Hinton et al., 2012) and language processing applications (Sutskever et al., 2014; Bahdanau et al., 2014). This broad success comes from a confluence of several factors. First, the creation of massive labeled datasets has allowed deep networks to demonstrate their advantages in expressiveness and scalability. The increase in computing power has also enabled training of far larger networks with more forgiving optimization dynamics (Choromanska et al., 2015). Additionally, architectures such as convolutional networks (LeCun et al., 1998) and long short-term memory networks (Hochreiter & Schmidhuber, 1997) have proven to be easier to optimize than classical feedforward and recurrent models. Finally, the success of deep networks

---

[*]First two authors contributed equally
[†]Work was done when author was at Google, Inc.

is also a result of the development of *simple* and *broadly applicable* learning techniques such as dropout (Srivastava et al., 2014), ReLUs (Nair & Hinton, 2010), gradient clipping (Pascanu et al., 2013; Graves, 2013), optimization algorithms and weight initialization strategies (Glorot & Bengio, 2010; Sutskever et al., 2013; He et al., 2015).

Recent work has aimed to push neural network learning into more challenging domains, such as question answering or program induction. These more complicated problems demand more complicated architectures (e.g. Graves et al. (2014); Sukhbaatar et al. (2015)), thereby posing new optimization challenges. While there is very active research in improving learning in deep feedforward and recurrent networks, such as layer-wise deep supervision (Lee et al., 2015), novel activation functions (Maas et al., 2013), initialization schemes (He et al., 2015), and cell architectures (Cho et al., 2014a; Yao et al., 2015), these are not always sufficient or applicable in networks with complex structure over the latent variables. In order to achieve good performance, researchers have reported the necessity of additional techniques such as explicit labeling of latent variables (Weston et al., 2014), relaxing weight-tying constraints (Kaiser & Sutskever, 2016), warmstarts (Peng et al., 2015), random restarts, and the removal of certain activation functions in early stages of training (Sukhbaatar et al., 2015).

The recurring theme is that commonly-used optimization techniques are not always sufficient to robustly optimize the models of interest. In this work, we explore a simple technique of adding annealed Gaussian noise to the gradient, which we find to be surprisingly effective in training deep neural networks with stochastic gradient descent. While there is a long tradition of adding random weight noise in neural networks, it has been under-explored in the optimization of modern deep architectures. Furthermore, although weight and gradient noise are equivalent when using standard SGD updates, the use of adaptive and momentum based stochastic optimizers such as Adam and AdaGrad (Duchi et al., 2011; Kingma & Ba, 2014) breaks this equivalence, allowing the noise to effectively adapt to the curvature of the optimization landscape. We find this property to be important when optimizing the most complex models.

While there exist theoretical and empirical results on the regularizing effects of conventional stochastic gradient descent, especially for the minimization of convex losses (Bousquet & Bottou, 2008), we find that in practice the added noise can actually help us achieve lower training loss by encouraging active exploration of parameter space. This exploration proves especially necessary and fruitful when optimizing neural network models containing many layers or complex latent structures. For neural network learning, it has long been known that the noise in the stochastic gradient can help to escape saddle points and local optima (Bottou, 1992). For this reason, neural network practitioners sometimes avoid overly-large mini-batch sizes to achieve the best results. We find that the Gaussian noise added in our technique is complementary to the noisy stochastic gradient, and a combination of Gaussian noise and tuned mini-batch sizes is necessary for the most complex models.

The main contribution of this work is to demonstrate the broad applicability of this simple method to the training of many complex modern neural architectures. To our knowledge, neither the exponentially decayed noise schedule nor the black box combination of injected gradient noise with adaptive optimizers have been used before in the training of deep networks. We consistently see improvements from Gaussian gradient noise when optimizing a wide variety of models, including very deep fully-connected networks, and special-purpose architectures for question answering and algorithm learning. For example, this method allows us to escape a poor initialization and successfully train a 20-layer rectifier network on MNIST with standard gradient descent. It also enables a 72% relative reduction in error in question answering, and doubles the number of accurate binary multiplication models learned across 7,000 random restarts. Gradient noise also possesses attractive robustness properties. We examine only two distinct settings of the noise variance hyperparameter in total across all experiments. We additionally observe that in cases where gradient noise fails to improve over other learning techniques, it rarely significantly hurts a models ability to generalize.

We hope that practitioners will see similar improvements in their own research by adding this simple technique, implementable in a single line of code, to their repertoire.

## 2 RELATED WORK

Adding random noise to the weights, inputs, or hidden units has been a known technique amongst neural network practitioners for many years (e.g. Murray & Edwards; An (1996)). However, the benefits of gradient noise have not been fully explored with modern deep networks nor combined with advanced stochastic optimization techniques, which allow the noise to take into account the geometry of the optimization problem and the statistical manifold.

Weight noise (Steijvers, 1996) and adaptive weight noise (Graves, 2011; Blundell et al., 2015), which usually maintains a Gaussian variational posterior over network weights, similarly aim to improve learning by added noise during training. In adaptive weight noise, an extra set of parameters for the variance must be maintained. This adaptation is different than our use of an adaptive optimizer, as it aims to capture an accurate estimate of uncertainty in the weights and not guide the exploration of parameter space. They differ from our proposed method in that the noise is not annealed and at convergence will be non-zero.

Similarly, the technique of dropout (Srivastava et al., 2014) randomly sets groups of hidden units to zero at train time to improve generalization in a manner similar to ensembling.

An annealed Gaussian gradient noise schedule was used to train the highly non-convex Stochastic Neighbor Embedding model in Hinton & Roweis (2002). The gradient noise schedule that we found to be most effective is very similar to the Stochastic Gradient Langevin Dynamics (SGLD) algorithm of Welling & Teh (2011), who use gradients with added noise to accelerate MCMC inference for logistic regression and independent component analysis models. This use of gradient information in MCMC sampling for machine learning to allow faster exploration of state space was previously proposed by Neal (2011). However, standard SGLD analysis does not allow for the use of adaptive optimizers or momentum, limiting the efficiency for very pathological optimization landscapes. Stochastic Gradient Riemannian Langevin Dynamics (Patterson & Teh, 2013) adapts the gradient and noise using the Fisher information matrix, effectively following trajectories along the same manifold as the natural gradient (Amari, 1998), but is applied only to models for which that matrix is tractable to estimate in closed form.

Various optimization techniques have been proposed to improve the training of neural networks. Most notable is the use of momentum (Polyak, 1964; Sutskever et al., 2013; Kingma & Ba, 2014) or adaptive learning rates (Duchi et al., 2011; Dean et al., 2012; Zeiler, 2012). These methods are normally developed to provide good convergence rates for the convex setting, and then heuristically applied to nonconvex problems. Similarly, batch normalization and related methods (Ioffe & Szegedy, 2015; Arpit et al., 2016; Salimans & Kingma, 2016), natural gradient descent (Amari, 1998; Desjardins et al., 2015), and K-FAC (Martens & Grosse, 2015) can all be seen as various preconditioning methods using approximations to the inverse Fisher information of the neural network. While there has been some difficulty in combining batch normalization-type algorithms with recurrent networks (Laurent et al., 2015), recent work has had success in this area (Cooijmans et al., 2016; Ba et al., 2016).

Injecting noise in the gradient can be combined with any of the above methods, and can be seen as a complementary technique especially suitable for nonconvex problems. By adding additional artificial stochasticity to the gradient, this technique allows the model more chances to escape local minima or saddle-points (see a similar argument in Bottou (1992)), or to traverse quickly through the "transient" plateau phase of early learning (see a similar analysis for momentum in Sutskever et al. (2013)). This is born out empirically in our observation that adding gradient noise can actually result in lower training loss. In this sense, we suspect adding gradient noise is similar to simulated annealing (Kirkpatrick et al., 1983) which exploits random noise to explore complex optimization landscapes. This can be contrasted with well-known benefits of stochastic gradient descent as a learning algorithm (Robbins & Monro, 1951; Bousquet & Bottou, 2008), where both theory and practice have shown that the noise induced by the stochastic process aids generalization by reducing overfitting.

Recently, there has been a surge in research examining the use of gradient and weight noise when training deep neural networks. Mobahi (2016) present an optimization technique for recurrent networks that applies an annealed Gaussian kernel smoothing method to the loss function, of which annealed weight noise is a Monte Carlo estimator. Li et al. (2016) present a version of SGLD that

incorporates both Gaussian noise and adaptively estimated learning rates (but no momentum term). Though significantly more complex than our proposed method, the most similar work is the *Santa* algorithm of Chen et al. (2016). Santa combines SGLD with adaptive learning rates and adaptive per-coordinate momentum parameters, and shows that the scheme can approach global optima of the objective function under certain assumptions.

## 3 METHOD

We consider a simple technique of adding time-dependent Gaussian noise to the gradient $g$ at every training step $t$:

$$g_t \leftarrow g_t + N(0, \sigma_t^2)$$

The gradient $g_t$ is then used to update the weights $\theta_t$ as if it were the original gradient of the loss function, and can be used with any stochastic optimization algorithm. Our experiments indicate that adding annealed Gaussian noise by decaying the variance often works better and more robustly than using fixed Gaussian noise (see Section 4.6). We use a schedule inspired from Welling & Teh (2011) in our experiments and take:

$$\sigma_t^2 = \frac{\eta}{(1+t)^\gamma} \tag{1}$$

We examine only 2 distinct noise hyperparameter configurations in our experiments, selecting $\eta$ from $\{0.01, 1.0\}$ and setting $\gamma = 0.55$ in all experiments. We believe this shows that annealed gradient noise is robust to minimal tuning. For example, in the experiments on Neural Programmer and Neural GPUs, we tried only a single configuration of noise parameters, simply setting $\eta = 1.0$ and tuning only the model hyperparameters as normal.

## 4 EXPERIMENTS

In the following experiments, we examine the effect of gradient noise on deep networks for MNIST digit classification, and consider a variety of complex neural network architectures: End-To-End Memory Networks (Sukhbaatar et al., 2015) and Neural Programmer (Neelakantan et al., 2016) for question answering, Neural Random Access Machines (Kurach et al., 2016) and Neural GPUs (Kaiser & Sutskever, 2016) for algorithm learning. The models and results are described as follows.

### 4.1 DEEP FULLY-CONNECTED NETWORKS

For our first set of experiments, we examine the impact of adding gradient noise when training a very deep fully-connected network on the MNIST handwritten digit classification dataset (LeCun et al., 1998). Our network is deep: it has 20 hidden layers, with each layer containing 50 hidden units, posing a significant optimization and generalization problem. We use the ReLU activation function (Nair & Hinton, 2010).

In this experiment, we train with SGD without momentum, using the fixed learning rates of 0.1 and 0.01. Unless otherwise specified, the weights of the network are initialized from a Gaussian with mean zero, and standard deviation of 0.1, which we call *Simple Init*. When adding gradient noise, we tried both settings of the variance detailed in Section 3, and found that decaying variance according to the schedule in Equation (1) with $\eta = 0.01$ worked best.

The results of our experiment are in Table 1. When trained from Simple Init we can see that adding noise to the gradient helps in achieving higher average and best accuracy over 20 runs using each learning rate for a total of 40 runs (Table 1, Experiment 1). We note that the average is closer to 50% because the small learning rate of 0.01 usually gives very slow convergence. We also try our approach on a more shallow network of 5 layers, but adding noise does not improve the training in that case.

Next, we experiment with clipping the gradients with two threshold values: 100 and 10 (Table 1, Experiment 2, and 3). Here, we find training with gradient noise is insensitive to the gradient

clipping values. By tuning the clipping threshold, it is possible to get comparable accuracy without noise for this problem.

In our fourth and fifth experiments (Table 1, Experiment 4), we use two analytically-derived ReLU initialization techniques (which we term *Good Init 1* and *2*) recently-proposed by Sussillo (2014) and He et al. (2015), and find that adding gradient noise does not help. Previous work has found that stochastic gradient descent with carefully tuned initialization, momentum, learning rate, and learning rate decay can optimize such extremely deep fully-connected ReLU networks (Srivastava et al., 2015). It would be harder to find such a robust initialization technique for the more complex heterogeneous architectures considered in later sections. Accordingly, we find in later experiments (e.g., Section 4.3) that random restarts and the use of a momentum-based optimizer like Adam are not sufficient to achieve the best results in the absence of added gradient noise.

To test how sensitive the methods are to poor initialization, in addition to the sub-optimal Simple Init, we run an experiment where all the weights in the neural network are initialized at zero. The results (Table 1, Experiment 5) show that if we do not add noise to the gradient, the networks fail to learn. If we add some noise, the networks can learn and reach 94.5% accuracy. While the pessimal performance of the noiseless model is unsurprising (initializing weights at 0 introduces symmetries that make gradient-descent impossible), it is interesting to note that gradient noise can overcome what is perhaps the canonical "bad initialization."

Experiment 1: Simple Init, No Gradient Clip

| Setting | Best Test Acc. | Avg. Test Acc. |
|---|---|---|
| No Noise | 89.9% | 43.1% |
| With Noise | 96.7% | 52.7% |
| No Noise + Dropout | 11.3% | 10.8% |

Experiment 2: Simple Init, Gradient Clip = 100

| No Noise | 90.0% | 46.3% |
|---|---|---|
| With Noise | 96.7% | 52.3% |

Experiment 3: Simple Init, Gradient Clip = 10

| No Noise | 95.7% | 51.6% |
|---|---|---|
| With Noise | 97.0% | 53.6% |

Experiment 4: Good Init 1 + Gradient Clip = 10

| No Noise | 97.4% | 92.1% |
|---|---|---|
| With Noise | 97.5% | 92.2% |

Experiment 5: Good Init 2 + Gradient Clip = 10

| No Noise | 97.4% | 91.7% |
|---|---|---|
| With Noise | 97.2% | 91.7% |

Experiment 6: Bad Init (Zero Init) + Gradient Clip = 10

| No Noise | 11.4% | 10.1% |
|---|---|---|
| With Noise | 94.5% | 49.7% |

Table 1: Average and best test accuracy on MNIST over 40 runs. Higher values are better.

In summary, these experiments show that if we are careful with initialization and gradient clipping values, it is possible to train a very deep fully-connected network without adding gradient noise. However, if the initialization is poor, optimization can be difficult, and adding noise to the gradient is a good mechanism to overcome the optimization difficulty. Additionally, the noise need not be heavily tuned and rarely decreases performance.

This set of results suggests that added gradient noise can be an effective mechanism for training complex networks. This is because it is more difficult to initialize the weights properly for these

architectures. In the following, we explore the training of more complex models such as End-To-End Memory Networks and Neural Programmer, whose initialization is less well studied.

## 4.2 END-TO-END MEMORY NETWORKS

We test added gradient noise for training End-To-End Memory Networks (Sukhbaatar et al., 2015), an approach for question answering using deep networks. Memory Networks have been demonstrated to perform well on a relatively challenging toy question answering problem (Weston et al., 2015).

In Memory Networks, the model has access to a context, a question, and is asked to predict an answer. Internally, the model has an attention mechanism which focuses on the right clue to answer the question. In the original formulation (Weston et al., 2015), Memory Networks were provided with additional supervision as to what pieces of context were necessary to answer the question. This was replaced in the End-To-End formulation by a latent attention mechanism implemented by a softmax over contexts. As this greatly complicates the learning problem, the authors implement a two-stage training procedure: First train the networks with a linear attention, then use those weights to warmstart the model with softmax attention.

In our experiments with Memory Networks, we use the same model hyperparameter settings as Sukhbaatar et al. (2015), and we try both settings of the variance detailed in Section 3, finding $\eta = 0.01$ worked best for this task. This noise is added to the gradient after clipping.

We set the number of training epochs to 200 because we would like to understand the behaviors of Memory Networks near convergence. We test the effect of gradient noise with the published two-stage training approach, and additionally with a one-stage approach where we train the networks with softmax attention and without warmstarting. Following the experimental protocol of Sukhbaatar et al. (2015), we take the model with lowest training error out of 10 random restarts. Results are reported in Table 2. We find some fluctuations during each run of the training, but the reported results reflect the typical gains obtained by adding random noise.

We find that warmstarting does indeed help the networks. In all cases, adding random noise to the gradient also helps the network both in terms of training errors and validation errors, and never hurts. Added noise, however, is especially helpful for the training of End-To-End Memory Networks without the warmstarting stage.

### One-Stage Training

| Setting | No Noise | With Noise |
|---|---|---|
| Train error: | 10.5% | 9.6% |
| Validation error: | 19.5% | 16.6% |

### Two-Stage Training

| | | |
|---|---|---|
| Train error: | 6.2% | 5.9% |
| Validation error: | 10.9% | 10.8% |

Table 2: The effects of adding gradient noise to End-to-End Memory Networks. Lower values are better.

## 4.3 NEURAL PROGRAMMER

Neural Programmer is a neural network architecture augmented with a small set of built-in arithmetic and logic operations that learns to induce latent programs. It is proposed for the task of question answering from tables (Neelakantan et al., 2016). Examples of operations on a table include the sum of a set of numbers, or the list of numbers greater than a particular value. Key to Neural Programmer is the use of "soft selection" to assign a probability distribution over the list of operations. This probability distribution weighs the result of each operation, and the cost function compares this weighted result to the ground truth. This soft selection, inspired by the soft attention mechanism of Bahdanau et al. (2014), allows for full differentiability of the model. Running the model for

several steps of selection allows the model to induce a complex program by chaining the operations, one after the other. At convergence, the soft selection tends to become peaky (hard selection). Figure 1 shows the architecture of Neural Programmer at a high level.

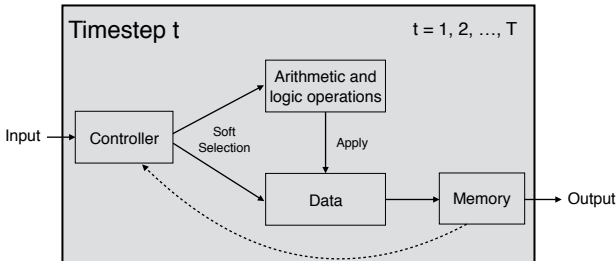

Figure 1: Neural Programmer, a neural network with built-in arithmetic and logic operations. At every time step, the controller selectes an operation and a data segment. Figure reproduced with permission from Neelakantan et al. (2016).

In a synthetic table comprehension task, Neural Programmer takes a question and a table (or database) as input and the goal is to predict the correct answer. To solve this task, the model has to induce a program and execute it on the table. A major challenge is that the supervision signal is in the form of the correct answer and not the program itself. The model runs for a fixed number of steps, and at each step selects a data segment and an operation to apply to the selected data segment. Soft selection is performed at training time so that the model is differentiable, while at test time hard selection is employed.

We examine only the noise configuration with $\eta = 1.0$, and add noise to the gradient after clipping, optimizing all other hyperparameters of the model. The model is optimized with Adam (Kingma & Ba, 2014), which combines momentum and adaptive learning rates.

For our first experiment, we train Neural Programmer to answer questions involving a single column of numbers. We use 72 different hyper-parameter configurations with and without adding annealed random noise to the gradients. We also run each of these experiments for 3 different random initializations of the model parameters and we find that only $1/216$ runs achieve 100% test accuracy without adding noise while $9/216$ runs achieve 100% accuracy when random noise is added. The 9 successful runs consisted of models initialized with all the three different random seeds, demonstrating robustness to initialization. We find that when using dropout (Srivastava et al., 2014) none of the 216 runs give 100% accuracy.

We consider a more difficult question answering task where tables have up to five columns containing numbers. We also experiment on a task containing one column of numbers and another column of text entries. Table 3 shows the performance of adding noise vs. no noise on Neural Programmer.

Question Answering Accuracy

| Setting | Dropout | No Noise | With Noise |
|---|---|---|---|
| Five columns | No | 95.3% | 98.7% |
| Text entries | No | 97.6% | 98.8% |
| Five columns | Yes | 97.4% | 99.2% |
| Text entries | Yes | 99.1% | 97.3% |

Table 3: The effects of adding random noise to the gradient on Neural Programmer. Higher values are better. Adding random noise to the gradient always helps the model. When the models are applied to these more complicated tasks than the single column experiment, using dropout and noise together seems to be beneficial in one case while using only one of them achieves the best result in the other case.

Figure 2 shows an example of the effect of adding random noise to the gradients in our experiment with 5 columns. The differences between the two models are much more pronounced than Table 3 indicates because that table reflects the results from the best hyperparameters. Figure 2 indicates a more typical training run.

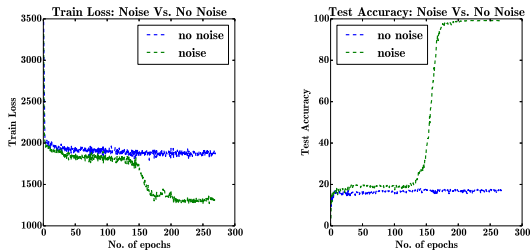

Figure 2: Noise Vs. No Noise in our experiment with 5 columns. The models trained with noise generalizes almost always better.

In all cases, we see that added gradient noise improves performance of Neural Programmer. Its performance when combined with or used instead of dropout is mixed depending on the problem, but the positive results indicate that it is worth attempting on a case-by-case basis.

## 4.4 NEURAL RANDOM ACCESS MACHINES

We now conduct experiments with Neural Random-Access Machines (NRAM) (Kurach et al., 2016). NRAM is a model for algorithm learning that can store data, and explicitly manipulate and derefer-ence pointers. NRAM consists of a neural network controller, memory, registers and a set of built-in operations. This is similar to the Neural Programmer in that it uses a controller network to com-pose built-in operations, but both reads and writes to an external memory. An operation can either read (a subset of) contents from the memory, write content to the memory or perform an arithmetic operation on either input registers or outputs from other operations. The controller runs for a fixed number of time steps. At every step, the model selects a "circuit" to be executed: both the operations and its inputs.

These selections are made using soft attention (Bahdanau et al., 2014) making the model end-to-end differentiable. NRAM uses an LSTM (Hochreiter & Schmidhuber, 1997) controller. Figure 3 gives an overview of the model.

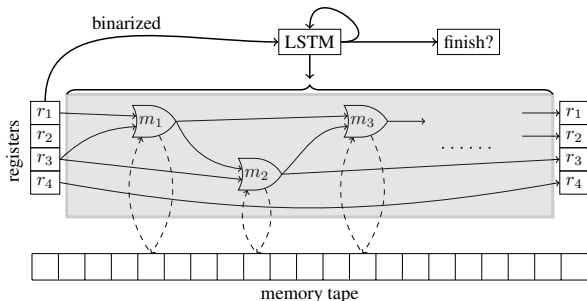

Figure 3: One timestep of the NRAM architecture with $R = 4$ registers and a memory tape. $m_1$, $m_2$ and $m_3$ are example operations built-in to the model. The operations can read and write from memory. At every time step, the LSTM controller softly selects the operation and its inputs. Figure reproduced with permission from Kurach et al. (2016).

For our experiment, we consider a problem of finding the $k$-th element's value in a linked list. The network is given a pointer to the head of the linked list, and has to find the value of the $k$-th element. Note that this is highly nontrivial because pointers and their values are stored at random locations in memory, so the model must learn to traverse a complex graph for $k$ steps.

Because of this complexity, training the NRAM architecture can be unstable, especially when the number of steps and operations is large. We once again experiment with the decaying noise schedule from Equation (1), setting $\eta = 0.01$. We run a large grid search over the model hyperparameters (detailed in Kurach et al. (2016)), and find the top 3 parameter settings separately for both noised

and un-noised models. For each model, for each of these 3 settings, we try 100 different random initializations and look at the percentage of runs that give $100\%$ accuracy across each one for training both with and without noise.

As in our experiments with Neural Programmer, we find that adding the noise after gradient clipping is crucial. This is likely because the effect of random noise is washed away when gradients become too large. For models trained with noise we observed much better reproduce rates, which are presented in Table 4. Although it is possible to train the model to achieve $100\%$ accuracy without noise, it is less robust across multiple random restarts, with over 10x as many initializations leading to a correct answer when using noise.

|  | No Noise | With Noise |
|---|---|---|
| Hyperparameter-1 | 1% | 5% |
| Hyperparameter-2 | 0% | 22% |
| Hyperparameter-3 | 3% | 7% |
| Average | 1.3% | 11.3% |

Table 4: Percentage of successful runs on the $k$-th element task. All tests were performed with the same set of 100 random initializations (seeds). Higher values are better.

## 4.5 CONVOLUTIONAL GATED RECURRENT NETWORKS (NEURAL GPUS)

Convolutional Gated Recurrent Networks (CGRN) or Neural GPUs (Kaiser & Sutskever, 2016) are a recently proposed model that is capable of learning arbitrary algorithms. CGRNs use a stack of convolution layers, unfolded with tied parameters like a recurrent network. The input data (usually a list of symbols) is first converted to a three dimensional tensor representation containing a sequence of embedded symbols in the first two dimensions, and zeros padding the next dimension. Then, multiple layers of modified convolution kernels are applied at each step. The modified kernel is a combination of convolution and Gated Recurrent Units (GRU) (Cho et al., 2014b). The use of convolution kernels allows computation to be applied in parallel across the input data, while the gating mechanism helps the gradient flow. The additional dimension of the tensor serves as a working memory while the repeated operations are applied at each layer. The output at the final layer is the predicted answer.

The key difference between Neural GPUs and other architectures for algorithmic tasks (e.g., Neural Turing Machines (Graves et al., 2014)) is that instead of using sequential data access, convolution kernels are applied in parallel across the input, enabling the use of very deep and wide models. The model is referred to as Neural GPU because the input data is accessed in parallel. Neural GPUs were shown to outperform previous sequential architectures for algorithm learning on tasks such as binary addition and multiplication, by being able to generalize from much shorter to longer data cases.

In our experiments, we use Neural GPUs for the task of binary multiplication. The input consists two concatenated sequences of binary digits separated by an operator token, and the goal is to multiply the given numbers. During training, the model is trained on 20-digit binary numbers while at test time, the task is to multiply 200-digit numbers. We add Gaussian noise with decaying variance according to the schedule in Equation (1) with $\eta = 1.0$, to the gradient after clipping. The model is optimized using Adam (Kingma & Ba, 2014).

Table 5 gives the results of a large-scale experiment using Neural GPUs with a 7290 grid search. The experiment shows that models trained with added gradient noise are more robust across many random initializations and parameter settings. As you can see, adding gradient noise both allows us to achieve the best performance, with the number of models with $< 1\%$ error over twice as large as without noise. But it also helps throughout, improving the robustness of training, with more models training to higher error rates as well. This experiment shows that the simple technique of added gradient noise is effective even in regimes where we can afford a very large numbers of random restarts.

| Setting | Error $< 1\%$ | $< 2\%$ | $< 3\%$ | $< 5\%$ |
|---------|---------------|---------|---------|---------|
| No Noise | 28 | 90 | 172 | 387 |
| With Noise | 58 | 159 | 282 | 570 |

Table 5: Number of successful runs on 7290 random trials. Higher values are better. The models are trained on length 20 and tested on length 200.

## 4.6 DISCUSSION

In this work we propose an annealed Gaussian gradient noise scheme for the optimization of complex neural networks. Our experiments show improvement from gradient noise on a variety of models. We conduct a small set of additional experiments below to examine the factors that make this technique successful, and report a failure mode.

**Annealed vs. fixed noise** We use a single fixed decay value $\gamma = 0.55$ when applying Equation (1) in our experiments, inspired by Stochastic Gradient Langevin Dynamics, and recommend it as a default. We conduct several experiments to determine the importance of annealed vs. fixed noise added to the gradient. We find that for the End2End model, similar results can be achieved with fixed noise values, however requiring significantly more tuning (compared to trying only two different values of $\eta$ in our experiments with annealed noise). We achieve nearly identical results on the End2End experiment using a fixed noise value of $\eta = 0.001$. We also experiment with fixed noise on the Neural Programmer and NRAM models, and find that they make a larger difference. For both models, we select fixed noise values log-uniformly from between $1e-4$ and $0.1$ and optimize the other hyperparameters. Using 216 runs per variance setting, the best Neural Programmer models without annealing can achieve equivalent errors to the annealed models. However, only 5/216 achieve the best error compared to 9/216 for the model using annealing. For NRAM, using 180 runs per setting, fixed noise never achieves the perfect error of 0 that is achieved by the annealed model. While annealing shows the most benefit with the most complex models, we generally recommend it as a robust default that requires less hyperparameter tuning than fixed noise.

**Gaussian noise vs. gradient stochasticity** We assert that gradient noise helps the model explore the optimization landscape, escaping saddle points and local minima. Analysis of SGD for neural networks suggests that the stochasticity of the gradient serves much the same purpose (Bottou, 1992). This suggests a strategy: add noise to the gradient by simply reducing the minibatch size, increasing the variance of the gradient estimator. While arguments based on SGLD and kernel smoothing provide evidence that the specific form of the Gaussian noise is important, we run a pair of small experiments. For both Neural Programmer and NRAM, we tried batch sizes of 10, 25, and 50 (50 being the value used in the best results). For NRAM, after 100 tasks at each batch size and no gradient noise, 2 tasks at batch size 50 converged to 0 error, 1 task at batch size 10, and none at batch size 25. For Neural Programmer, over 216 experiments at each batch size we see none of the models without gradient noise converge to the best error. These results are far worse than our results using added noise, indicating that merely lowering the batch size does not introduce the same sort of helpful stochasticity.

**Gradient noise vs. weight noise** While weight noise is relatively well-known, it is not equivalent to gradient noise in the case of adaptive or momentum-based optimizers, which effectively adapt the noise to the curvature of the optimization landscape. Both Neural Programmer and NRAM are greatly helped in training by the use of the Adam algorithm for optimization. We find here, using the same experimental setup as when examining annealed vs. fixed noise, that the models fail to learn when adding noise directly to the weights. Even when using starting noise rates as low as $1e-6$, with the usual annealing schedule, the models fail to train significantly, achieving 57% error for NRAM and 68% for Neural Programmer at the lowest. Importantly, these noise rates are on the same order as the adaptive learning rates. This indicates that the issue is not just the noise scale, but that the very poor conditioning of the loss functions makes it necessary to adapt the noise. Similar concerns motivated the development of very recent algorithms for preconditioned SGLD in the Bayesian setting (Li et al., 2016).

**Negative results**    While we see improvements on a large number of neural network architectures, we note a case where gradient noise does not improve over standard SGD. We conduct language modeling experiments on the Penn Treebank (Marcus et al., 1993), using the experimental setup and architecture from Zaremba et al. (2014). We report results using a 200-unit LSTM with dropout, but observe a similar lack of improvement from gradient noise when using models without dropout. We try the two proposed noise rates from Section (method) and find the best results using $\eta = 0.01$ are slightly worse than the noiseless model, achieving a perplexity of 98 rather than 95. By further lowering the noise parameter to $\eta = 0.001$ we are able to achieve the same perplexity as the baseline, but do not see improvement. While adding gradient noise does not help in this case, it is simple to try and does not significantly hurt the results.

## 5    CONCLUSION

In this paper, we demonstrate the effectiveness of adding noise to the gradient when training deep neural networks. We find that adding noise to the gradient helps optimization and generalization of complicated neural networks and is compatible with and complementary to other stochastic optimization methods. We suspect that the effects are pronounced for complex models because they have many saddle points.

We believe that this surprisingly simple yet effective idea, essentially a single line of code, should be in the toolset of neural network practitioners when facing issues with training neural networks.

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
