# Peer review of "Adding Gradient Noise Improves Learning for Very Deep Networks"

_ICLR 2017 — rejected_

[Public Comment · Tim Cooijmans · 15 Dec 2016]
**Use with batch-normalized recurrent networks**

This work was done quite a while ago, and in the meantime optimization of recurrent neural networks has been improved by various batch normalization-like schemes. Have you tried combining the technique with recurrent batch normalization[1], layer normalization[2] or weight normalization[3]? I'd be very curious to see whether the technique still brings improvements for normalized networks.

[1]

[Official Review · AnonReviewer3 · rating 7 · confidence 5 · 15 Dec 2016]
**Review: Adding Gradient Noise Improves Learning for Very Deep Networks**

The authors propose to add noise to the gradients computed while optimizing deep neural networks with stochastic gradient based methods. They show results multiple data sets which indicate that the method can counteract bad parameter initialization and that it can be especially beneficial for training more complicated architectures.

The method is tested on a multitude of different tasks and architectures. The results would be more convincing if they would be accompanied by confidence intervals but I understand that some of the experiments must have taken very long to run. I like that the results include both situations in which the gradient noise helps a lot and situations in which it doesn’t seem to add much to the other optimization or initialization tools employed. The quantity of the experiments and the variety of the models provide quite convincing evidence that the effect of the gradient noise generalizes to many settings. The results were not always that convincing. In Section 4.2, the method only helped significantly when a sub-optimal training scheme was used, for example. The results on MNIST are not very good compared to the state-of-the-art. Since the method is so simple, I was hoping to see more theoretical arguments for its usefulness. That said, the experimental investigations into the importance of the annealing procedure, the comparison with the effect of gradient stochasticity and the comparison with weight noise, provide some additional insight.

The paper is well written and cites relevant prior work. The proposed method is described clearly and concisely, which is to be expected given its simplicity. 

The proposed idea is not very original. As the authors acknowledge, very similar algorithms have been used for training and it is pretty much identical to simulating Langevin dynamics but with the goal of finding a single optimum in mind rather than approximating an expected value. The work is the evaluation of an old tool in a new era where models have become bigger and more complex.

Despite the lack of novelty of the method, I do think that the results are valuable. The method is so easy to implement and seems to be so useful for complicated model which are hard to initialize, that it is important for others in the field to know about it. I suspect many people will at least try the method. The variety of the architectures and tasks for which the method was useful suggests that many people may also add it to their repertoire of optimization tricks. 


Pros:
* The idea is easy to implement.
* The method is evaluated on a variety of tasks and for very different models.
* Some interesting experiments which compare the method with similar approaches and investigate the importance of the annealing scheme.
* The paper is well-written.


Cons:
* The idea is not very original.
* There is no clear theoretical motivation of analysis.
* Not all the results are convincing.

[Official Review · AnonReviewer1 · rating 4 · confidence 5 · 17 Dec 2016 (modified: 21 Jan 2017)]

This paper presents a simple method of adding gradient noise to improve the training of deep neural networks. This paper first appeared on arXiv over a year ago and while there have been many innovations in the area of improving the training of deep neural networks in tha time (batch normalization for RNNs, layer normalization, normalization propagation, etc.) this paper does not mention or compare to these methods. 

In particular, the authors state "However, recent work on applying batch normalization to recurrent networks (Laurent et al., 2015) has not shown promise in improving generalization ability for recur- rent architectures, which are the focus of this work." This statement is simply incorrect and was thoroughly explored in, e.g. Cooijmans et al. (2016) that establish that batch normalization is effective for RNNs.

The proposed method itself is extremely simple and is similar to numerous training strategies that have previously been advocated in the literature. As a result the contribution would be incremental at best and could be significant with sufficiently strong empirical results supporting this particular variant. However, as discussed above there are now multiple training strategies and algorithms in the literature that are not empirically compared.

Unfortunately, this paper is now fairly seriously out of date. It would not be appropriate to publish this at ICLR 2017.

[Official Review · AnonReviewer2 · rating 4 · confidence 4 · 21 Dec 2016]
**better than no noise, but lack of comparison with results in the literature**

The authors consider a simple optimization technique consisting of adding gradient noise with a specific schedule. They test their method on a number of recently proposed neural networks for simulating computer logic (end-to-end memory network, neural programmer, neural random access machines).

On these networks, the question of optimization has so far not been studied as extensively as for more standard networks. A study specific to this class of models is therefore welcome. Results consistently show better optimization properties from adding noise in the training procedure.

One issue with the paper is that it is not clear whether the proposed optimization strategy permits to learn actually good models, or simply better than those that do not use noise. A comparison to results obtained in the literature would be desirable.

For example, in the MNIST experiments of Section 4.1, the optimization procedure reaches in the most favorable scenario an average accuracy level of approximately 92%, which is still far from having actually learned an interesting problem representation (a linear model would probably reach similar accuracy). I understand that the architecture is specially designed to be difficult to optimize (20 layers of 50 HUs), but it would have been more interesting to consider a scenario where depth is actually beneficial for solving the problem.

[Final Decision · Program Chairs · 06 Feb 2017]
**ICLR committee final decision**

This paper presents a simple heuristic for training deep nets: add noise to the gradients at time t with variance $\eta/(1+t)^\gamma$. There is a battery of experimental tests, representing a large amount of computer time. However, these do not compare to any similar ideas, some of which are theoretically motivated. For example the paper identifies SANTA as the closest related work, but there is no comparison.
 
 We encourage the authors to address these outstanding issues and to resubmit.